# Mapping the energy level alignment at donor/acceptor interfaces in non-fullerene organic solar cells

Xian'e Li [1✉], Qilun Zhang [1], Jianwei Yu[2], Ye Xu[3], Rui Zhang[2], Chuanfei Wang [1], Huotian Zhang [2], Simone Fabiano [1], Xianjie Liu[1], Jianhui Hou [3], Feng Gao [2] & Mats Fahlman [1✉]

Energy level alignment (ELA) at donor (D) -acceptor (A) heterojunctions is essential for understanding the charge generation and recombination process in organic photovoltaic devices. However, the ELA at the D-A interfaces is largely underdetermined, resulting in debates on the fundamental operating mechanisms of high-efficiency non-fullerene organic solar cells. Here, we systematically investigate ELA and its depth-dependent variation of a range of donor/non-fullerene-acceptor interfaces by fabricating and characterizing D-A quasi bilayers and planar bilayers. In contrast to previous assumptions, we observe significant vacuum level (VL) shifts existing at the D-A interfaces, which are demonstrated to be abrupt, extending over only 1–2 layers at the heterojunctions, and are attributed to interface dipoles induced by D-A electrostatic potential differences. The VL shifts result in reduced interfacial energetic offsets and increased charge transfer (CT) state energies which reconcile the conflicting observations of large energy level offsets inferred from neat films and large CT energies of donor - non-fullerene-acceptor systems.

[1] Laboratory of Organic Electronics, Department of Science and Technology (ITN), Linköping University, Norrköping SE-60174, Sweden. [2] Biomolecular and Organic Electronics, Department of Physics, Chemistry and Biology (IFM), Linköping University, Linköping SE-58183, Sweden. [3] Beijing National Laboratory for Molecular Sciences, State Key Laboratory of Polymer Physics and Chemistry, Institute of Chemistry, Chinese Academy of Sciences, 100190 Beijing, China. ✉email: xiane.li@liu.se; mats.fahlman@liu.se

In organic solar cells (OSCs), both charge generation and charge recombination occur at the donor (D)–acceptor (A) interfaces. Therefore, the energy level alignment (ELA) at D–A interfaces is a key parameter for a fundamental understanding of OSCs. For example, in fullerene-acceptor-based OSCs (FA-OSCs), an energy offset of over 0.3 eV is usually observed in efficient devices. This offset, determined by the energy difference between the highest occupied ($\Delta_{HOMO}$) or the lowest unoccupied ($\Delta_{LUMO}$) molecular orbitals of the donor and acceptor, was commonly believed to provide the necessary driving force for efficient charge separation[1–5].

Contrary to this common belief in FA-OSCs, a wide range of non-fullerene-acceptor-based OSCs (NFA-OSCs) shows high efficiencies under negligible or even zero HOMO or LUMO offsets[6–15]. The energy offsets are usually either estimated from the optical measurements by determining the energy difference between singlet excitons and charge-transfer states or estimated from cyclic voltammetry (CV) measurements, assuming that the vacuum level alignment at the D–A interfaces provides a rough estimation of the D–A interface ELA. The contrast between FA-OSCs and NFA-OSCs indicates that different regimes of device physics and photophysics are reached in NFA-OSCs. However, some recent reports challenge efficient charge generation upon zero energy offsets in NFA-OSCs, and claim that a decent HOMO offset is still necessary for efficient charge generation in NFA-OSCs[16–21]. For example, Karuthedath et al. found that a HOMO offset of 0.5 eV is required for efficient NFA-OSCs. In this case, the HOMO levels are determined by ultraviolet photoelectron spectroscopy (UPS), and quadrupole field effects bending the energy levels at the D–A interfaces are assumed (Fig. 1a).

These contradicting observations partially result from the challenges in experimentally determining the ELA at the D–A interfaces. In principle, in order to capture both the evolution of the energy levels and the change in vacuum level in the D–A interface region, it is required to measure these properties "layer-by-layer"[22]. The donors and NFAs cannot be vacuum deposited into thin films due to their large size, and spin-coating uniform monolayers on top of each other and without intermixing are equally daunting. Most previous reports hence assumed no vacuum level shifts, an assumption that is not proven in NFA-OSCs.

In this work, we investigate the ELA at well-defined D–A interfaces by developing both quasi and planar bilayer heterojunctions based on a set of high-performing D–A pairs. Work function (WF) and ionization potential (IP) evolutions with film thickness are mapped out through building up multilayer films with monolayer precision while avoiding interface intermixing using the Langmuir-Schäfer (LS) technique. In contrast to the previous assumptions of constant VL at the D–A interfaces, we observe a large interface VL shift (Fig. 1b) extending 1–2 layers from the interface. At the same time, the IP stays constant at the neat material value from the first monolayer at the heterojunction regardless of D–A combination. By exploring an extended set of donor and acceptor materials, we find that an enhanced VL shift occurs for all D–A systems featuring significant intramolecular charge transfer and a large $\Delta_{HOMO}$ prior to contact. The cause of the enhanced VL shift is tentatively assigned to intermolecular charge transfer induced by strong electrostatic coupling between the donors and acceptors at the interface. The ELA landscapes mapped out from the D–A interfaces in our work rationalize the co-existence of efficient charge separation and small voltage losses in NFA-OSCs, providing insights for further development of high-efficiency NFA-OSCs.

## Results

**ELA in quasi-bilayer D–A films.** ELA diagrams of donors and acceptors found in the literature are usually derived from HOMO and LUMO energies measured from neat materials assuming VL alignment at the D–A interface, which often does not reflect the actual ELA due to the formation of interface dipole steps that shifts the VL[23,24]. To explore the ELA and in particular if VL shifts occur at NFA-OSC D–A interfaces, we carefully follow the changes of WF and IP of films by UPS during the layer-by-layer deposition process. As shown in Supplementary Fig. 1, one can define the VL of each layer and obtain the VL shifts ($\Delta$VL) or WF changes ($\Delta$WF) at the interface with the Fermi level ($E_F$) serving as energy reference, once the sample is contacted to the

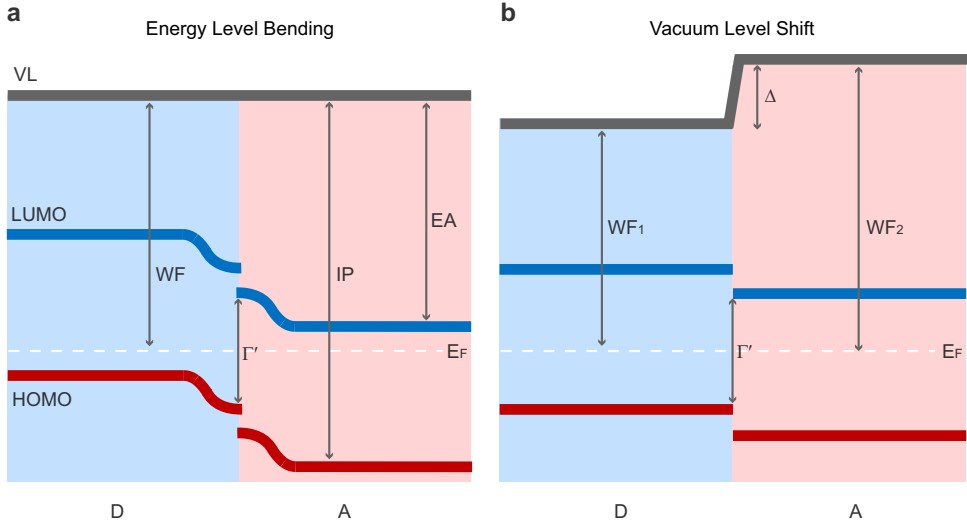

**Fig. 1 Schematic energy level diagram of donor (D)-acceptor (A) interfaces. a** Energy level bending (ELB) scenario and **b** vacuum level (VL) shift scenario. The energy difference between the VL and the highest occupied molecular orbitals (HOMO) level is denoted as ionization potential (IP). The energy difference between VL and the lowest unoccupied molecular orbitals (LUMO) level is denoted as electron affinity (EA). The photovoltaic gap ($\Gamma'$) could be enhanced either by bending of energy levels at the interface (ELB scenario) or by interface dipole ($\Delta$) induced shifting of the vacuum level (VL shift scenario). In the VL shift scenario, work function (WF) of each phase is denoted as $WF_1$ and $WF_2$, respectively. Fermi level (white dashed line) is denoted as $E_F$.

spectrometer and grounded. Here, two donors (PBDB-T, PM6) and two acceptors (ITIC, IT4F) are selected as model materials, as they yield high power conversion efficiencies (PCEs) in the corresponding OSCs and share, respectively, similar molecular backbones (Supplementary Fig. 2), yet have different energy levels. To avoid the influence of substrate intrinsic dipoles on the determination of ELA in D–A heterojunctions, and to mimic the electrode contact in a common device structure, two non-pre-polarized substrates, Au and $AlO_x$ (Al with a native oxide layer) are employed as high-WF substrates for the D(bottom)/A(top) films and low-WF substrates for A(bottom)/D(top) films, respectively.

Quasi-bilayer D–A films were fabricated by spin-coating (SC) the donor as the bottom layer on UVO-treated Au substrates (WF = 5.2–5.7 eV) and then spin-coating a NFA top layer using an orthogonal solvent, which yields a D–A bilayer that is close to a planar heterojunction[25]. UPS measurement was carried out to obtain the WF and the IP of the film before and after coating the NFA top layer. As can be seen in Supplementary Fig. 3, large WF shifts of 0.60–0.71 eV are found in all spin-coated bilayer films compared with their corresponded bottom donor layers, while the measured IPs are the same as those in neat ITIC (IP = 5.74 ± 0.1 eV) and IT4F (IP = 5.79 ± 0.1 eV) films.

UPS being a surface-sensitive technique, the measured IP is obtained almost exclusively from the topmost molecular layer, so the IP for the layers at or near the D–A interface is unknown. Similarly, though the change in WF or the corresponding VL shift is captured, its vertical location and spatial extension are not evident from the measurement due to the multilayer thickness of the NFA top layer. Furthermore, angle-dependent near-edge X-ray absorption fine structure (NEXAFS) spectroscopy data of the spin-coated donor and acceptor films (Supplementary Fig. 4, Supplementary Table 1) show that the donor polymers lack any preferential orientation. In contrast, ITIC films are preferentially face-on oriented, in agreement with reported results[26], whereas the IT4F films are more disordered (Supplementary Note 1). The D–A interfaces here, and in real devices, thus consist of a mixture of face-on/face-on, edge-on/face-on, and edge-on/edge-on oriented heterojunctions, and it is not obvious which type dominates the observed interface ELA.

**ELA in planar bilayer D–A Films**. To gain more insight into the ELA, we employ the Langmuir-Schäfer technique to build up planar D–A bilayer films monolayer-by-monolayer yielding abrupt (non-intermixed) interfaces and well-defined molecular orientations. This technique makes it possible to map out the ELA, including IP and WF evolution for each molecular layer in the vertical stack[27].

We fabricate homogenous monolayer films and multilayer films of the donors and acceptors by depositing closely packed LS films onto Au or $AlO_x$ substrate. A combination of surface pressure, atomic force microscope (AFM), and NEXAFS measurements (Supplementary Tables 1 and 2, Supplementary Figs. 4 and 5, Supplementary Note 1) shows that the LS films all feature edge-on molecular orientation, though the NEXAFS data suggest that IT4F films are slightly less ordered compared to the ITIC films.

The WF and IP are measured layer by layer when deposited either with donor monolayers on Au substrates followed by acceptor monolayers (Au/D/A), or acceptor monolayers on $AlO_x$ followed by donor monolayers ($AlO_x$/A/D). The UPS spectra are shown in Supplementary Fig. 6 and the corresponding energy level evolution is depicted in Supplementary Fig. 7. As summarized in Fig. 2, at the metal-organic interface, the WF of NFA films on $AlO_x$ shifts due to a charge-transfer induced

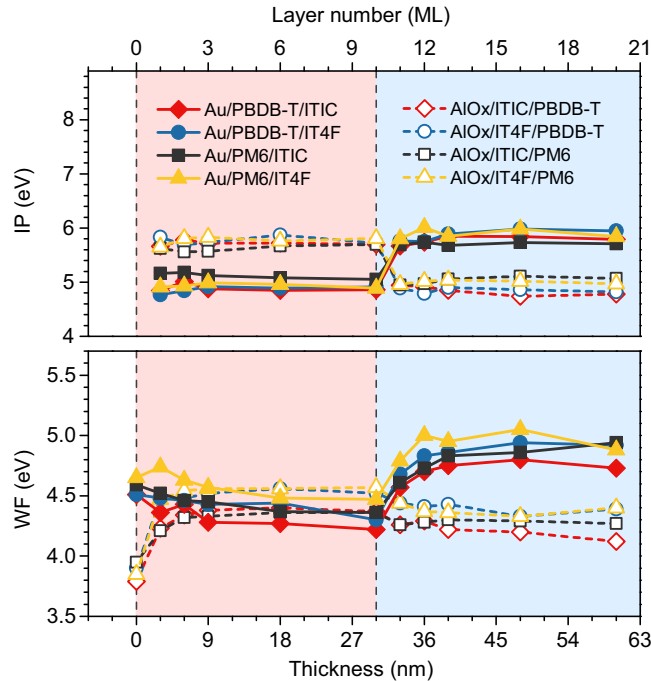

**Fig. 2 Mapping the energy level alignment at the donor (D)–acceptor (A) interfaces in monolayer precision.** IP (ionization potential) and WF (work function) evolution of Au/D/A (solid lines) and $AlO_x$/A/D (dashed lines) heterojunctions as a function of D, A layer number (top axis) or the corresponding film thickness (bottom axis, presented by taking 3 nm as the average thickness of a monolayer). Upon the donor and acceptor stacking order, the substrate is in white, bottom phase is in red, and top phase is in blue.

interface dipole and remains constant after 1–2 monolayers, which is the predicted behavior[24,28]. The donor on Au interface ($WF_{Au}$ = 4.5–4.6 eV) also follows the established model, displaying a small interface dipole step or VL alignment as their pinning energies dictate (see Supplementary Fig. 8). There is no change in the IP going from the monolayer (ML) adjacent to the metal substrate to the 10 ML thick bulk equivalent film for any of the donors or NFAs. At the D–A interface for both the Au/D/A and $AlO_x$/A/D planar bilayer heterojunctions, we again see abrupt shifts of the WF (completed in 1–2 ML): a large increase for the NFA top layer cases and a relatively smaller decrease for the donor top layer cases, see Supplementary Table 3. There is no change in the IP going from the monolayer at the organic heterojunction to the 10 ML thick bulk equivalent top film for any of the donors or NFAs. The ELA of these systems thus features an abrupt VL shift at the interface and no bending of the HOMO, from which we infer that the LUMO is not undergoing bending either. This result rules out the ELB at interface but well supports the VL shift scenario.

A similar set of interfaces were prepared where the bottom NFA or donor layer was spin-coated onto the metal substrate, whereupon the corresponding top layers were added by LS deposition. In this way we obtain abrupt D–A interfaces with the top LS films featuring edge-on orientation (IT4F LS films are less ordered) and the bottom spin-coated films featuring both face-on and edge-on orientations. The ELA displays near-identical behavior as the LS/LS films of edge-on/edge-on orientation with the VL shifts are of similar size and direction (see a summary of ΔWF values for four model materials in Table 1). Again, there is no evidence of bending of the HOMO. For all cases, it is evident that films with donors at the bottom and acceptors on top all

**Table 1 Summary of the ΔWF values for donor (D)-acceptor (A) interfaces based on different fabrication methods.**

| Fabrication method | SC/SC | SC/SS | LS/LS | LS/LS | SC/LS | SC/LS |
|---|---|---|---|---|---|---|
| D-A interfaces | $\Delta WF_{D/A}$ | $\Delta WF_{A/D}$ | $\Delta WF_{D/A}$ | $\Delta WF_{A/D}$ | $\Delta WF_{D/A}$ | $\Delta WF_{A/D}$ |
| PBDB-T:ITIC | +0.60 | −0.33 | +0.48 | −0.10 | +0.48 | −0.32 |
| PBDB-T:IT4F | +0.65 | −0.36 | +0.41 | −0.18 | +0.39 | −0.33 |
| PM6:ITIC | +0.60 | −0.16 | +0.32 | −0.25 | +0.49 | −0.35 |
| PM6:IT4F | +0.71 | −0.20 | +0.57 | −0.16 | +0.42 | −0.34 |

WF difference between D-A bilayers and neat bottom layers is denoted as ΔWF, in which $\Delta WF_{D/A} = WF_{Au/D/A} − WF_{Au/D}$, $\Delta WF_{A/D} = WF_{AlOx/A/D} − WF_{AlOx/A}$. The bilayer films are prepared by spin-coating/spin-coating (SC/SC), spin-coating/surface-spreading (SC/SS), Langmuir-Schäfer/Langmuir-Schäfer (LS/LS), and spin-coating/Langmuir-Schäfer (SC/LS) methods respectively. All values are shown in units of eV.

show VL upshifts (positive ΔWF values) at the D–A interfaces, while those with acceptors at the bottom and donors on top always show VL downshifts (negative ΔWF values). An interface dipole pointing from acceptor to donor side hence exists at all investigated donor–NFAs interfaces, whereas its size differs in D/A and A/D films. Generally, D/A films show a larger dipole size than A/D films regardless of the molecular orientation. Since the bilayer films are fabricated from different methods which may result in different molecular packing, we tentatively attribute this to the lower reorganization energy for the small acceptor molecules compared to the large bulky donor polymers to find an optimal position at the interface. The SC/SC films exhibit a relatively larger interface dipole step size than films made from other methods, which can be due to the face-on/face-on oriented interfaces that are absent or at least less in the LS/LS and SC/LS bilayers. The relatively flexible movements of donors or acceptors assisted by the solvent molecules during the spin-coating process additionally may enable conversion of edge-on/face-on to face-on/face-on, enhancing the VL shift. Finally, to test the influence of substrate type (metal vs non-metal), interface dipoles are measured in D/A and A/D bilayer films prepared on two non-metallic substrates, ITO/PEDOT:PSS and ITO/ZnO, respectively. The values obtained are in good agreement with the bilayer films on Au and $AlO_x$ substrates (Supplementary Table 4, Supplementary Fig. 9).

**Interface dipole-induced vacuum level shift enhances the energy of interfacial states**. We have demonstrated significant and abrupt VL shifts for all bilayer structures and that there is no bending of the energy levels, enabling us to provide a more reliable ELA diagram at in situ D–A interfaces. Figure 3 gives the ELA information at D–A interfaces before and after contact for all combinations of the four model materials, using the VL shift measured from SC/SC films which we deem more likely to reflect the dominant situation in devices (face-on/face-on orientation). Before contact or the VL shift correction, all combinations show very large interfacial energetic offsets, with $\Delta_{HOMO}$ or $\Delta_{LUMO}$ values larger than 0.6 eV, and the photovoltaic gap Γ (difference between neat donor HOMO and neat acceptor LUMO) smaller than 1.2 eV. Such large energy level offsets and small interfacial state energies contradict the experimental facts that most high-efficiency NFA-OSCs exhibit high charge-transfer state energies ($E_{CT}$)[29,30], open-circuit voltage ($V_{OC}$), and barrierless free charge generation process[31]. However, after contact and the VL shift correction, the resulting ELA displays interfacial energetic offsets that are shrunk to near zero while the photovoltaic gaps are significantly increased (see Table 2).

To correlate the corrected ELA diagrams with real devices, OSC devices based on the four model materials are fabricated. Their current density–voltage (J–V) characteristics are tested and presented in Supplementary Fig. 10 and Supplementary Table 5.

In addition, external quantum efficiency ($EQE_{PV}$) and electroluminescence (EL) measurements are carried out for each device, and the CT energies are determined by fitting the reduced $EQE_{PV}$ and EL spectra according to the approach by Vandewal et al.[32] (Supplementary Fig. 11). As summarized in Table 2, Γ′, the interfacial state energy after VL shift increases to the range of 1.43–1.81 eV and comes much closer to the interfacial CT state energy $E_{CT}$ measured in the device, which exactly confirms each other species. A linear cascaded drop from the photovoltaic gap to CT energy and then to final $V_{OC}$ is correctly obtained using Γ′ as a reference, while this trend is missing entirely in the ELA obtained without the VL shift correction (Fig. 4).

The directly measured D–A interface ELA rationalizes many of the contradicting observations in NFA-OSCs. The abrupt and large VL shift at the heterojunctions provides a small to near-zero $\Delta_{HOMO}$ and hence a photovoltaic gap close to the optical gap. The direction of the interface dipole enhances the CT exciton dissociation into free charges and decreases recombination[33,34]. Our results thus largely confirm the findings of Karuthedath and co-workers, with the important distinction that there is no ELB at the heterojunctions but instead an interface dipole that converts a large $\Delta_{HOMO}$ based on neat film values into a near-zero $\Delta_{HOMO}$ at the D–A interfaces.

**Universal interface dipoles in organic–organic heterojunctions**. The VL shifts found at the D–A interfaces for the studied systems are significantly larger than what is expected from integer charge transfer (ICT)[24,35,36] induced interface dipole steps (see Supplementary Fig. 12). To investigate if other organic semiconductor materials undergo similar enhanced VL shifts at D–A interfaces, we study additional types of donors and acceptors including (1) the newly developed high-efficiency Y-series NFAs (Y6, Y11), (2) another ultra-narrow bandgap NFA series (IEICO, IEICO-4F), (3) fullerene acceptors ($C_{60}$, $PC_{70}BM$), (4) a new type donor of high PCE (D18), (5) old type donors (TQ1, P3HT), and (6) the wide-bandgap polymer PFO. Their energy levels and pinning energies measured in neat films on inorganic and organic conducting substrates follow the expected behavior and are presented in Supplementary Table 6 and Supplementary Fig. 13. We measure the interface dipoles at the various D–A interfaces (Supplementary Table 7), compare them with the estimated dipoles of respective D–A interfaces according to the ICT model (Supplementary Table 8), and obtain differences referred to as enhanced VL shifts. The relationship between the enhanced VL shift measured at D–A interfaces and the HOMO offset obtained from neat films of the respective materials is depicted in Fig. 5.

We generally find enhanced D–A interface VL shifts in bilayers based on NFAs, largely absent in the FAs systems. In particular, significantly enhanced interface VL shifts are observed in systems with both donors and acceptors featuring A-D alternating structures (donor: PBDB-T, PM6, D18, TQ1, acceptor: ITIC,

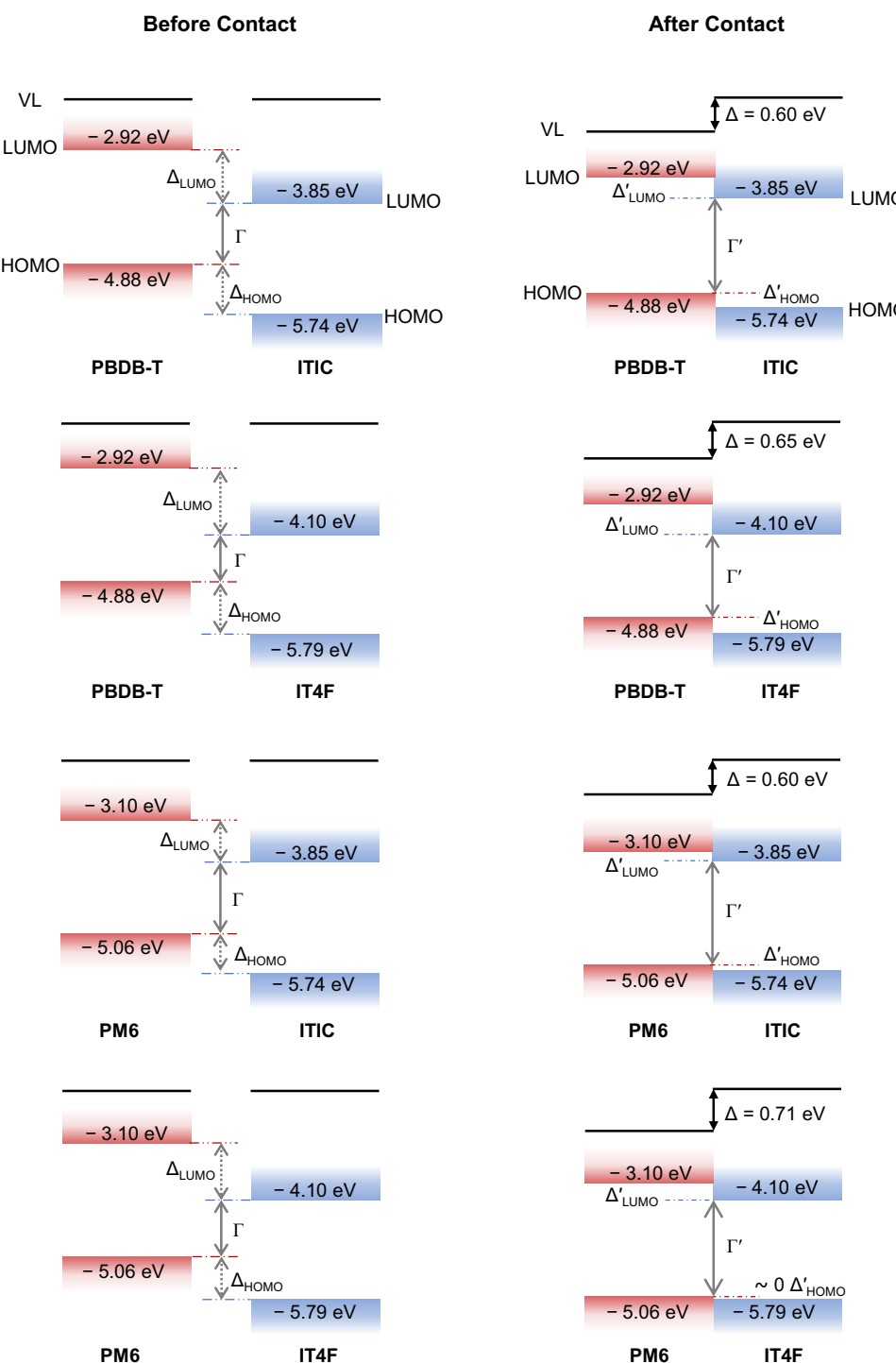

**Fig. 3 Energy level diagrams of D–A interfaces before and after contact.** All HOMO (highest occupied molecular orbitals) levels are measured by UPS, and LUMO (lowest unoccupied molecular orbitals) levels are obtained from literature and measured by inverse photoelectron spectroscopy (IPES)[21], except for that of PBDB-T, which is obtained from the CV method[29]. HOMO difference between donor and acceptor before (after) conatct is denoted as $\Delta_{HOMO}$ ($\Delta'_{HOMO}$). LUMO difference between donor and acceptor before (after) conatct is denoetd as $\Delta_{LUMO}$ ($\Delta'_{LUMO}$). Photovoltaic gap at the D–A interface before (after) contact is denoted as $\Gamma(\Gamma')$. $\Delta$ is the interface dipole inducing the vacuum level (VL) shift.

IT4F, Y6, Y11). On the contrary, no enhanced VL shift (>0.1 eV) is found for NFAs combined with donors lacking the push-pull structure, with the exception of the P3HT/ITIC combination. This suggests that significant intramolecular charge transfer is a necessary but not sufficient prerequisite for obtaining a large enhanced VL shift. We also observe that the interfaces featuring donors and acceptors with small HOMO offsets based on the neat

film IPs (i.e., before contact) only show small or negligible dipoles. This trend is clearly shown when we divide the data points in Fig. 5 into different quadrants defined by two solid lines, enhanced VL shift = 0.1 eV and HOMO offset = 0.3 eV, respectively. Systems with large enhanced VL shifts (>0.1 eV) can only be found in the quadrant ① where their HOMO offsets are also prominent (>0.3 eV). The donor and acceptor systems in this

**Table 2 Summary of the energetic offset $\Delta_{HOMO}$ ($\Delta'_{HOMO}$), $\Delta_{LUMO}$ ($\Delta'_{LUMO}$), photovoltaic gap $\Gamma$ ($\Gamma'$) between donor (D) and acceptor (A) before (after) VL shift correction, as well as the charge-transfer state energy ($E_{CT}$) and open-circuit voltage ($V_{OC}$) values measured in devices.**

| D(bottom)/A(top) | $\Delta_{HOMO}$ (eV) | $\Delta'_{HOMO}$ (eV) | $\Delta_{LUMO}$ (eV) | $\Delta'_{LUMO}$ (eV) | $\Gamma$ (eV) | $\Gamma'$ (eV) | $E_{CT}$ (eV) | $V_{OC}$ (V) |
|---|---|---|---|---|---|---|---|---|
| PBDB-T/ITIC | 0.86 | 0.26 | 0.93 | 0.33 | 1.03 | 1.63 | 1.50 | 0.89 |
| PBDB-T/IT4F | 0.91 | 0.26 | 1.18 | 0.53 | 0.78 | 1.43 | 1.31 | 0.69 |
| PM6/ITIC | 0.68 | 0.08 | 0.75 | 0.15 | 1.21 | 1.81 | 1.59 | 1.00 |
| PM6/IT4F | 0.73 | 0.02 | 1.00 | 0.29 | 0.96 | 1.67 | 1.46 | 0.82 |

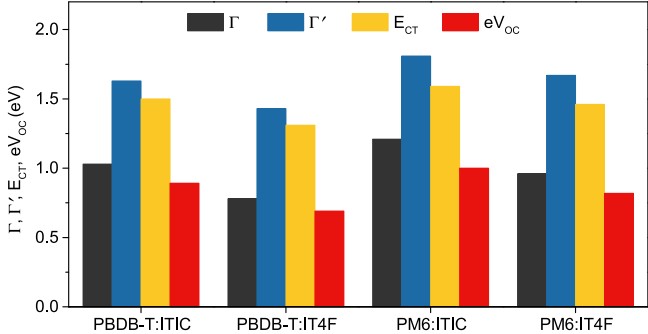

**Fig. 4 Vacuum level shift enhances the energy of interfacial states.**
Comparison of the charge-transfer state energy ($E_{CT}$) and open-circuit voltage ($V_{OC}$) values measured in devices with the photovoltaic gaps $\Gamma$ ($\Gamma'$) before (after) VL shift correction.

region share a common feature of significant intramolecular charge transfer. No D–A combination is found in the quadrant ② where the enhanced VL shift size is larger than the HOMO offset. In the low HOMO offset region (quadrant ③) where HOMO offset <0.3 eV, no enhanced VL shift occurs regardless of molecular types. The systems composed of $C_{60}$ are all located in the lower part of the quadrant ④ with negligible VL shifts. The enhanced VL shift is then influenced by two factors jointly: HOMO offset between corresponding neat films and the degree of intramolecular charge transfer of the donor and acceptor.

## Discussion

The ELA at organic heterojunctions typically follows the ICT model, and as noted, all of the materials studied display the expected behavior when deposited onto inorganic and organic conducting substrates. However, the ICT model assumes the absence of intrinsic electric fields at the interface and no orbital hybridization. As we see no significant change in the HOMO energy of the donors or acceptors at the heterojunctions, significant hybridization is unlikely but not ruled out completely. Hence, the enhanced VL shifts obtained for the D–A systems featuring strong intramolecular charge transfer in both the donor and acceptor molecules and a significant $\Delta_{HOMO}$ may have their origin in charge transfer across the D–A interface driven by an intrinsic interfacial electric field. Significant intramolecular charge transfer will lead to large variations in molecular electrostatic potential (ESP) and hence increased attraction between donor and acceptor molecules at an interface. The increased electrostatic interaction can guide the donor–acceptor molecular ordering at the interface to maximize the interfacial electric field[37], and a large $\Delta_{HOMO}$ would make it highly possible that a net charge reorganization occurs, with the hole density on the donor and electron density on the acceptor.

In order to examine this hypothesis, ESP calculations are carried out on built-up molecular models of PBDB-T, PM6, ITIC,

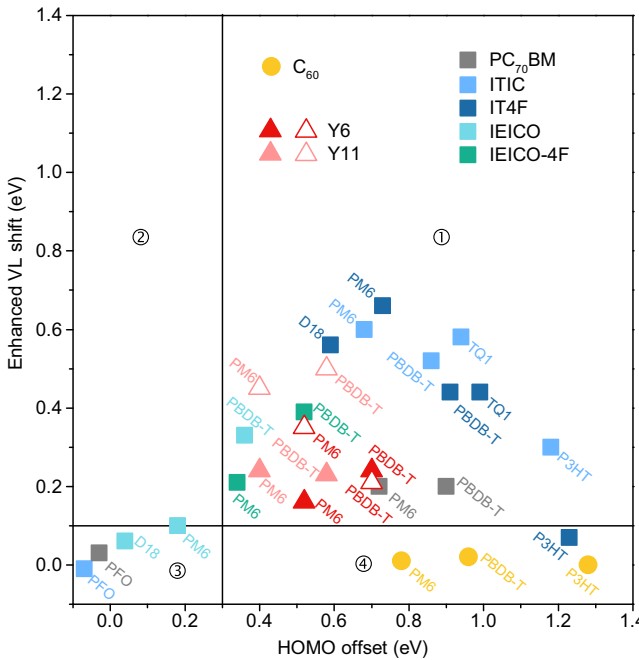

**Fig. 5 Enhanced vacuum level (VL) shifts versus HOMO offsets measured from neat films of the respective materials.** Different bilayer fabrication methods are clarified by the shapes of data points: squares for SC/SC, solid triangles for crosslinked-SC/SC, open triangles for SC/LS and circles for SC/evaporation, respectively. SC, LS, crosslinked-SC is for spin-coating, Langmuir-Schäfer, and spin-coating followed by crosslinking method, correspondingly.

and IT4F. The resulting ESP distributions of the four molecular models are mapped based on their molecular van de Waals surfaces[30]. As shown in Supplementary Fig. 14, the donors show mainly negative potential areas where yellow and red signify averaged ESP values of −76.39 and −17.36 meV, respectively. In contrast, most areas of acceptors show more positive potentials where cyan and blue correspond to averaged ESP values of 73.63 and 138.49 meV, respectively. To calculate the interface dipole induced between these molecules that have alternating D–A segments, molecular models of D units from donors (BDT-T from PBDB-T, BDT-TF from PM6) and A units from acceptors (IC unit from ITIC, IC-2F unit from IT4F) are built up with planar molecular geometries, and their calculated ESP distributions are displayed in Supplementary Fig. 15. The interface dipoles between D and A units are calculated with a face-to-face distance of 3.5 Å, see Supplementary Fig. 16. All the D–A unit pairs show consistent dipoles pointing from A unit to D unit, with dipole size of around 0.7 or 0.9 Debye. The assumption here is that the D–A interface comprises the densely packed dipole layer with a dipole density of $\mu_0/A$ where $\mu_0$ is the dipole moment

**Table 3 Summary of the calculated dipole dimensions for donor (D)–acceptor (A) interfaces of face-on/face-on orientation, calculated vacuum level potential steps ($\Delta VL_{cal.}$), and the ones measured from UPS experiments ($\Delta VL_{exp.}$).**

| D-A interfaces | Dipole dimension | $\Delta VL_{cal.}$ | $\Delta VL_{exp.}$ |
|---|---|---|---|
| PBDB-T:ITIC | −0.9326 Debye | 0.58 eV | 0.60 eV |
| PBDB-T:IT4F | −0.9275 Debye | 0.58 eV | 0.65 eV |
| PM6:ITIC | −0.7402 Debye | 0.46 eV | 0.60 eV |
| PM6:IT4F | −0.7292 Debye | 0.45 eV | 0.71 eV |

created by the isolated D–A unit pair, and $A$ is the corresponding overlapped molecular area. The vacuum level shift (ΔVL) induced by the collective electrostatic effects of the dipole layer could be calculated by

$$\Delta VL = -\frac{q_e \boldsymbol{\mu}_0}{\varepsilon_0 \varepsilon_r A} \qquad (1)$$

where $q_e$, $\varepsilon_0$, $\varepsilon_r$ correspond to the charge of an electron, vacuum permittivity, and relative permittivity, respectively[38]. Taking the overlapped area as 20 Å$^2$ and the relative permittivity as 3, the calculated ΔVL ($\Delta VL_{cal.}$) is shown in Table 3, in good agreement with the measured VL shifts ($\Delta VL_{exp.}$) from SC/SC films with the donor as the bottom layer.

The consistency of the dipole direction with its positive pole next to the donor side and negative pole next to the NFA side explains why, in the UPS measurements, there are always WF upshifts when the NFA is at the top and downshifts when the donor is at the top. Supplementary Fig. 17 shows the electron and hole density distributions at ground states based on the same D–A unit packing models, where significant charge transfer is observed in all systems. The calculations support the scenario where interface dipole causing the enhanced VL shift results from the transfer of electron density driven by an interfacial electrostatic field obtained from molecular ESP.

In summary, we provide a direct measurement of the ELA at a real D–A interface for NFA-based systems by building up both quasi and planar bilayers made from different methods, enabling different molecular orientations of preferentially face-on/face-on, face-on/edge-on, and edge-on/edge-on, as well as more randomly ordered interfaces. We find that large abrupt VL shifts occur at the D–A interfaces, while no energy level bending is evident, resulting in decreased $\Delta_{HOMO}$ that approaches near zero for some D–A combinations. The ELA diagrams after VL shift correction match well with the observations in corresponding NFA-OSCs, such as large photovoltaic gap, large CT energy, and small or negligible interfacial $\Delta_{HOMO}$. An extended investigation enables us to generalize that VL shifts occur for systems where the donor and acceptor feature significant intramolecular charge transfer (so-called push-pull systems) and where the $\Delta_{HOMO}$ obtained from neat film values is large (>0.3 eV). We propose that the enhanced VL shifts have an origin in ESP differences between the donor and acceptor materials, guiding the intermolecular alignment at the interface and inducing interfacial charge transfer. Our findings reconcile contradicting observations in the literature and provide insights into the fundamental understanding of operational mechanisms in NFA-OSCs.

## Methods

**Materials and film fabrication**. PBDB-T, PM6, P3HT, TQ1, PFO, Y6, ITIC, IT4F, PC$_{70}$BM, C$_{60}$ were obtained from 1-Material Inc. D18, IEICO, IEICO-4F, and Y11 were provided by Solarmer Materials Inc. The films for the pinning energy measurements were spin-coated from chloroform (CF) on various conductive substrates.

The conductive substrates chosen to provide a broad range of the work function were: Al/AlO$_x$ (WF$_{SUB}$ = 3.4–3.9 eV), indium tin oxide (ITO)/ZnO (WF$_{SUB}$ = 3.9–4.1 eV), Si/SiO$_x$ (WF$_{SUB}$ = 4.1–4.7 eV), ITO as-received or UV/ozone (UVO) -treated (WF$_{SUB}$ = 4.5, or 4.7–5.0 eV), PEDOT:PSS (WF$_{SUB}$ = 5.0–5.1 eV), gold (Au) exposed to air or UVO-treated (WF$_{SUB}$ = 4.5–4.6 eV, or 5.2–5.7 eV). All inorganic substrates were cleaned by sonication for 10 min in demineralized (DI) water, acetone, and isopropyl in sequence prior to use. All spin-coated (SC/SC) bilayer films were prepared by spin-coating donor from chlorobenzene (CB) (10 mg mL$^{-1}$) as the bottom layer, and then spin-coating acceptor from dichloromethane (DCM) (6 mg mL$^{-1}$) as the top layer. For the D–A bilayers with serious intermixing through this SC/SC method, the bottom layer was crosslinked before spin-coating the top layer (crosslinked-SC/SC). The all Langmuir-Schäfer (LS) deposition (LS/LS) bilayer films were layer-by-layer deposited by LS technique (KSV NIMA Instruments) on Au or Al/AlO$_x$ substrates in a cleanroom. Before deposition, materials were dissolved in CF with a concentration of 1 mg mL$^{-1}$, and then were diluted into a solution of 0.1 mg mL$^{-1}$. The diluted solution was randomly dispensed onto a pure deionized (DI) water subphase by a microsyringe. After solvent evaporation, the LS films were compressed continuously by two barriers at a rate of 5 mm min$^{-1}$ during which the surface pressure was monitored by Wilhelmy-plate method. The deposition of polymer layers was carried out by approaching the substrate horizontally to the air/water interface at a surface pressure of 25 mN m$^{-1}$, while small molecule layers were deposited at 30 mN m$^{-1}$. After deposition, the LS films were dried in a vacuum oven at 50 °C overnight. The SC/LS films were prepared by firstly spin-coating donor (acceptor) film at the bottom and then depositing three more Langmuir-Schäfer layers of the acceptor (donor) on the top. The spin-coating/surface-spreading (SC/SS) bilayer films were made by firstly spin-coating acceptor as the bottom layer, and then depositing the donor film formed on DI water surface through surface-spreading[39] method during which donor solution of 10 mg mL$^{-1}$ in CB was adopted.

**Ultraviolet photoelectron spectroscopy (UPS)**. UPS measurements were performed in a UHV surface analysis system including an entry chamber (base pressure ≈ 1 × 10$^{-7}$ mbar), a preparation chamber (≈ 8 × 10$^{-9}$ mbar), and an analysis chamber (≈ 2 × 10$^{-10}$ mbar). Samples were measured in the analysis chamber using monochromatized He I light ($h\nu$ = 21.22 eV) with the total energy resolution of 0.1 eV calibrated by the Fermi level of Ar$^+$ ion sputter-cleaned gold foil. The work function of organic films was derived from the secondary-electron cutoff edge and the vertical ionization potential (IP) was obtained from the frontier edge of the valence band spectrum. Radiation damage from the light source on the organic films was carefully examined and no damage was detected. All measurements were carried out in dark.

**Atomic force microscope (AFM)**. AFM measurements were performed using a Dimension 3100 microscope equipped with a Nanoscope IV controller (Bruker-Nano). Commercial silicon cantilevers with a nominal spring constant of 40 N m$^{-1}$ and with a resonance frequency in the 150–300 kHz range were used for morphological characterization in tapping mode. All images were recorded under ambient conditions.

**Near-edge X-ray absorption fine structure (NEXAFS) spectroscopy**. Measurements were performed at the FlexPES (Flexible Photoelectron Spectroscopy) beamline at MAX IV synchrotron radiation facility in Lund, Sweden, with horizontally linear polarized light in the energy range of 40–1500 eV. The defocused beam spot size at sample is ca. 2 × 1 mm to minimize any possible radiation damage. The energy resolution is about 20 meV at photon energies close to the C K-edge. Angle-dependent NEXAFS spectra were collected in both partial electron yield (PEY) and total electron yield (TEY) detection modes with multichannel plate and sample drain current, respectively.

**Device fabrication**. The inverted structure of ITO/ZnO/active layer (100 ± 5 nm) /MoO$_3$/Ag was applied for device fabrication. Patterned ITO (Lumtec) substrates were cleaned with deionized water, acetone, and isopropanol in an ultrasonic bath for 10 min each, followed by 15 min microwave plasma treatment at 200 W. Then ZnO nanoparticles (Avantama N-10) dispersed in isopropanol were filtered with a PTFE filter (0.45 μm) and spin-coated onto the cleaned ITO at 3600 rpm under ambient conditions, followed by the thermal annealing at 120 °C for 30 min. Donor (PBDB-T or PM6) and acceptor (ITIC or IT4F) were dissolved in chlorobenzene (CB) to a total concentration of 20 mg mL$^{-1}$ with 1:1 mass ratio and 0.5% DIO (v/v, DIO/ CB) as an additive, and the solution was stirred for 3 h inside the glovebox prior to use. Next, the blend was spin-coated onto the ZnO substrate with a speed of 2500–3000 rpm to get a photoactive layer of 100 nm thickness, followed by thermally annealed at 100 °C for 10 min. Finally, 10 nm MoO$_3$ (anode interlayer) and 100 nm Ag (top electrode) were deposited in sequence under a 10$^{-6}$–10$^{-7}$ mbar in a vacuum. The device area was determined from the top contact mask and layout of ITO substrate, and the resulting active area was 0.047 cm$^2$.

**Current density–voltage characteristics (J–V)**. J–V characteristics were recorded on the encapsulated devices in the air by a Keithley 2400 Source Meter (measured in the forward direction with a scan step of 0.04 V s$^{-1}$ at room temperature) under

the illumination of a 470 nm LED with an intensity equivalent to 1000 W m$^{-2}$ after spectral mismatch correction. The light intensity for the J–V measurements was calibrated with a reference Si cell (VLSI standards SN 10510-0524 certified by National Renewable Energy Laboratory).

**Electroluminescence (EL) and external quantum efficiency (EQE$_{EL}$) measurements**. EL spectra were collected in the dark while keeping the objective near the sample. Keithley 2400 Source Meter was used as the voltage supplier and Newton EM-CCD Si array was applied as the detector (−45 °C) equipped with a Shamrock SR-303i spectrograph from Andor Tech. EQE values were obtained from an in-house-built system in which Hamamatsu silicon photodiode 1010B was used as the light source. Keithley 2400 Source Meter was used to apply voltage and record the injected current, and Keithley 485 Picoammeter was applied to record the emitted light intensity according to the generated photocurrent.

**Electrostatic potential (ESP) simulation**. The molecular ESP provides the potential distribution around the molecule which can be observed physically by X-ray diffraction or derived computationally from the wave function[37]. The ESP at the point **r** in molecular surroundings V(**r**), is given by Eq. (2) based on the basic Coulomb's law including the contribution from both nuclei and electrons, where $\rho(\mathbf{r})$ is the electronic density function and $-e\rho(\mathbf{r})d\mathbf{r}$ is the electronic charge in each volume element d**r**, $Z_Ae$ is the charge on nucleus A at location $\mathbf{R}_A$, $|\mathbf{R}_A-\mathbf{r}|$ and $|\mathbf{r}'-\mathbf{r}|$ represent the distance of each charge element from **r**, $\varepsilon_0$ is the permittivity of free space. The magnitude and sign of V(**r**) are the same as the interaction energy of the system with a unit positive point charge placed at $\mathbf{r}$[40].

$$V(\mathbf{r}) = \frac{1}{4\pi\varepsilon_0}\left[\sum_A \frac{Z_Ae}{|\mathbf{R}_A - \mathbf{r}|} - e\int\frac{\rho(\mathbf{r}')d\mathbf{r}'}{|\mathbf{r}' - \mathbf{r}|}\right] \tag{2}$$

The ESP distribution is derived from the wave function calculated by Gaussian 09 on the calculation level of B3LYP/6-31G(d,p). The electron density analysis is carried out on Multiwfn[41].

**Reporting summary**. Further information on research design is available in the Nature Research Reporting Summary linked to this article.

## Data availability
The authors declare that the main data supporting the findings of this study are available within the paper and its Supplementary Information files. Source data are provided with this paper.

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

## Acknowledgements

We thank the financial support from the Swedish Research Council (project grants no. 2016-05498, 2016-05990, 2020-04538, and 2018-06048), the Swedish Energy Agency (grant. no. 45411-1), and by the Swedish Government Strategic Research Area in Materials Science on Functional Materials at Linköping University (Faculty Grant SFO Mat LiU no. 2009 00971). Q.Z. acknowledges support from the Wallenberg Wood Science Center (WWSC). F.G. acknowledges the Stiftelsen för Strategisk Forskning through a Future Research Leader program (FFL18-0322). We acknowledge MAX IV Laboratory for time on Beamline FlexPES under Proposal 20200292. Research conducted at MAX IV, a Swedish national user facility, is supported by the Swedish Research council under contract 2018-07152, the Swedish Governmental Agency for Innovation Systems under contract 2018-04969, and Formas under contract 2019-02496. We thank Dr. Alexei Preobrajenski's kind assistance with the NEXAFS experiments. X.L. thanks Dr. Jiquan Wu for the kind teaching and training on the Langmuir-Schäfer film deposition.

## Author contributions

X.L. conceived the study, prepared Langmuir-Schäfer and spin-coated films, performed AFM, UPS, NEXAFS measurements, data analysis, and prepared the first draft of the manuscript. Q.Z. prepared the surface-spreading films, performed NEXAFS measurements, part of the UPS measurements, and contributed to routine project discussion. J.Y. prepared and characterized the OSC devices, performed $EQE_{EL}$ measurements and data analysis. Y.X. performed the electrostatic potential simulation and calculated the dipole size between D, A unit. R.Z. prepared the bilayer films based on the crosslinked bottom layer. C.W. contributed to data analysis and manuscript preparation. H.Z. determined the CT energies from $EQE_{EL}$ spectrum. S.F. supervised the Langmuir-Schäfer film preparation and characterization. X.Liu supervised the UPS, NEXAFS measurements and contributed to the development and maintenance of UPS setup. J.H. supervised the electrostatic potential simulation. F.G. supervised the device preparation, characterization, CT energy determination and participated in the manuscript preparation. M.F. conceived the study, supervised the project and manuscript preparation. All authors contributed to the results discussion and final version of manuscript.

## Funding

## Competing interests

The authors declare no competing interests.
