## [Peer Review File · Nature Communications]

Reviewer comments, initial review -

Reviewer #1 (Remarks to the Author):

The authors study energy level changes, specifically vacuum level shifts, at organic donor/acceptor interfaces. Using UPS on well-defined planar DA bilayers with varying donor/acceptor thickness or molecular monolayer numbers they demonstrate that abrupt changes of the vacuum level occur at the interface spatially limited to no more than 1-2 acceptor monolayers. The abrupt change of the vacuum level at the interface modifies the interfacial energy level alignment such that energy offsets are reduced and concomitantly the diagonal bandgap between donor and acceptor is increased. The work provides insight into the currently debated observation of efficient charge separation in non-fullerene acceptor organic PV systems despite small energy offsets and supports recent work showing that sizeable IP offsets are required to ensure efficient exciton-to-charge conversion. The mechanism of charge generation in non-fullerene acceptor organic bulk heterojunctions is a matter of ongoing discussion in the OPV community, and precise insight into the interfacial energetics in NFA-based systems is largely missing, since experimentally difficult to determine. The authors demonstrate quite convincingly that contrary to the common notion that the vacuum level remains unchanged at the DA interface, it does abruptly change in NFA-based systems over a very narrow spatial region equivalent to a monolayer of molecules. In my opinion, this is a quite important observation, since it adds to a solution of the recent controversy about the minimum energy offsets needed in NFA-based, which still ensure efficient charge separation. It becomes clearer, why offsets on excess of 0.5 eV are required. Importantly, the authors also demonstrate by studying a broad set of donor and acceptor materials, under which conditions significant vacuum level shifts can be expected, which allows developing more precise design rules of novel materials. While the current version of the manuscript is already at an acceptable level, at least I could not identify any major flaws, the data seems to support the conclusion quite well, and the manuscript is also very well written, it would have been interesting to see if the 'corrected' HOMO/LUMO offsets determined by the authors show any correlation to the measured device performance, specifically the charge generation efficiency which could be approximated by the internal quantum efficiency of the devices. Establishing such a correlation and a better understanding of the extend of the VL shift depending on the donor/acceptor chemical structures could help developing more precise design rules. In that regard, the conclusion of the manuscript remains quite vague, since the occurrence of significant VL shifts is only linked to the push-pull type structure of molecules and IP differences in excess of 0.3eV.

Reviewer #2 (Remarks to the Author):

The manuscript by Li et al. presents the investigation of energy level alignment at interface of polymer donor and non-fullerene acceptor in organic solar cells. The interfacial energy alignment of the active layer of organic solar cells plays an important role on the exciton dissociation and charge transfer across the interface. Some recent works reported on the efficient charge generation for very small or even negligible band offsets in non-fullerene acceptor based organic solar cells, in contrary to the common requirement of moderate band offsets to drive the efficient exciton dissociation and charge transfer across the interface in the fullerene based organic solar cells. The underlying driving mechanism of effective charge transfer in non-fullerene acceptor based organic

solar cells with negligible band offset is yet unclear and demands in-depth studies. In addition, determining the precise band alignment and offset of the hetero-interface is the prerequisite for establishing the model of driving force of effective charge generation. From these perspective, the topic of the manuscript is very interesting and important.

However, I find the evidence of abrupt vacuum level shift near the interface, which is one of the key conclusions of the manuscript and the precondition of further discussion, is merely based on the work function measured by the ultraviolet photoelectron spectroscopy (UPS). As the authors indicated, the UPS is a surface sensitive technique. Thus, the change in the work function in the planar bilayer system cannot simply represent the vacuum energy shift. At least, the authors need to give strong evidence to support such a claim.

Moreover, it is still not clear throughout the manuscript why the formation of interface dipole can drive effective exciton dissociation and charge generation in the case of negligible band offsets.

Because of the above two major issues, I cannot recommend the publication of the manuscript in its present form.

Response to Reviewers' Comments on Manuscript NCOMMS-21-26681-T

We thank the reviewers for helpful suggestions on improving this manuscript. We have carefully read the comments and have answered all the questions accordingly as detailed below.

Responses to Reviewer #1:

Comments to the Author:

The authors study energy level changes, specifically vacuum level shifts, at organic donor/acceptor interfaces. Using UPS on well-defined planar DA bilayers with varying donor/acceptor thickness or molecular monolayer numbers they demonstrate that abrupt changes of the vacuum level occur at the interface spatially limited to no more than 1-2 acceptor monolayers. The abrupt change of the vacuum level at the interface modifies the interfacial energy level alignment such that energy offsets are reduced and concomitantly the diagonal bandgap between donor and acceptor is increased. The work provides insight into the currently debated observation of efficient charge separation in non-fullerene acceptor organic PV systems despite small energy offsets and supports recent work showing that sizeable IP offsets are required to ensure efficient exciton-to-charge conversion.

The mechanism of charge generation in non-fullerene acceptor organic bulk heterojunctions is a matter of ongoing discussion in the OPV community, and precise insight into the interfacial energetics in NFA-based systems is largely missing, since experimentally difficult to determine. The authors demonstrate quite convincingly that contrary to the common notion that the vacuum level remains unchanged at the DA interface, it does abruptly change in NFA-based systems over a very narrow spatial region equivalent to a monolayer of molecules. In my opinion, this is a quite important observation, since it adds to a solution of the recent controversy about the minimum energy offsets needed in NFA-based, which still ensure efficient charge separation. It becomes clearer, why offsets on excess of 0.5 eV are required. Importantly, the authors also demonstrate by studying a broad set of donor and acceptor materials, under which conditions significant vacuum level shifts can be expected,

which allows developing more precise design rules of novel materials.

While the current version of the manuscript is already at an acceptable level, at least I could not identify any major flaws, the data seems to support the conclusion quite well, and the manuscript is also very well written, it would have been interesting to see if the 'corrected' HOMO/LUMO offsets determined by the authors show any correlation to the measured device performance, specifically the charge generation efficiency which could be approximated by the internal quantum efficiency of the devices. Establishing such a correlation and a better understanding of the extend of the VL shift depending on the donor/acceptor chemical structures could help developing more precise design rules. In that regard, the conclusion of the manuscript remains quite vague, since the occurrence of significant VL shifts is only linked to the push-pull type structure of molecules and IP differences in excess of 0.3eV.

Reply (R): We sincerely thank the reviewer for his/her interest and attention on our work. Regarding to the nice suggestions or comments, we have answered them point by point as below.

Question (Q) 1. While the current version of the manuscript is already at an acceptable level, at least I could not identify any major flaws, the data seems to support the conclusion quite well, and the manuscript is also very well written, it would have been interesting to see if the ‘corrected’ HOMO/LUMO offsets determined by the authors show any correlation to the measured device performance, specifically the charge generation efficiency which could be approximated by the internal quantum efficiency of the devices. Establishing such a correlation and a better understanding of the extend of the VL shift depending on the donor/acceptor chemical structures could help developing more precise design rules. In that regard, the conclusion of the manuscript remains quite vague, since the occurrence of significant VL shifts is only linked to the push-pull type structure of molecules and IP differences in excess of 0.3eV

RI: We agree with the reviewer that it is an interesting topic to correlate the corrected energy level offsets with the measured device performance, and to further explore the charge generation mechanism in these NFA-based OSCs. This is beyond the scope of this manuscript, however, as we focused on mapping the energy level alignment at donor/acceptor interfaces in NFA-based OSCs with layer-by-layer accuracy. Indeed, by comparing our results with device data in literature we can see good agreement and that the vacuum level shifts provide the necessary adjustment of the energy levels to obtain near-zero-loss in the efficient systems we studied. A true correlation would require that we build up devices using our LS-fabrication technique, which in turn will require lamination-based devices, a very challenging task. Nevertheless, we are following the reviewer’s advice and are pursuing this topic for a future manuscript. The reviewer is also correct about the vagueness about the origin of the significant VL shift. We believe that our data firmly establishes not only the precise energy level alignment for a large set of donor-acceptor combinations but also gives guidelines for when the “extra” large VL shift will occur. Our results are also in line with reference 21 (S. Karuthedath, et al), where we now can explain how the ELA assumed in this paper actually occurs. The physical explanation of the mechanism yielding these VL shifts is indeed less certain, but we offer our best estimate.

Responses to Reviewer #2:

Comments to the Author:

The manuscript by Li et al. presents the investigation of energy level alignment at interface of polymer donor and non-fullerene acceptor in organic solar cells. The interfacial energy alignment of the active layer of organic solar cells plays an important role on the exciton dissociation and charge transfer across the interface. Some recent works reported on the efficient charge generation for very small or even negligible band offsets in non-fullerene acceptor based organic solar cells, in contrary to the common requirement of moderate band offsets to drive the efficient exciton dissociation and charge transfer across the interface in the fullerene based organic solar cells. The underlying driving mechanism of effective charge transfer in non-fullerene acceptor based organic solar cells with negligible band offset is yet unclear and demands in-depth studies. In addition, determining the precise band alignment and offset of the hetero-interface is the prerequisite for establishing the model of driving force of effective charge generation. From these perspective, the topic of the manuscript is very interesting and important.

However, I find the evidence of abrupt vacuum level shift near the interface, which is one of the key conclusions of the manuscript and the precondition of further discussion, is merely based on the work function measured by the ultraviolet photoelectron spectroscopy (UPS). As the authors indicated, the UPS is a surface sensitive technique. Thus, the change in the work

function in the planar bilayer system cannot simply represent the vacuum energy shift. At least, the authors need to give strong evidence to support such a claim.

Moreover, it is still not clear throughout the manuscript why the formation of interface dipole can drive effective exciton dissociation and charge generation in the case of negligible band offsets.

Because of the above two major issues, I cannot recommend the publication of the manuscript in its present form.

Reply (R): Sincerely thanks for the reviewer's interest and careful examination on our manuscript. We have answered your questions point by point as below.

Question (Q) 1. However, I find the evidence of abrupt vacuum level shift near the interface, which is one of the key conclusions of the manuscript and the precondition of further discussion, is merely based on the work function measured by the ultraviolet photoelectron spectroscopy (UPS). As the authors indicated, the UPS is a surface sensitive technique. Thus, the change in the work function in the planar bilayer system cannot simply represent the vacuum energy shift. At least, the authors need to give strong evidence to support such a claim.

RI: Yes, UPS is a surface sensitive technique, because only the electrons ionized from topmost molecular layers can survive from the inelastic scattering in film and finally contribute to most of the spectrum signal. But this does not mean that the measured work function changes in bilayer systems only reflect the work function changes in topmost layers. Instead, UPS is a powerful technique and has been widely used to follow the possible VL shift at both metal-organic and organic-organic interfaces (*Adv. Mater.* **11**, 605–625, 1999; *Org. Electron.* **11**, 212–217, 2010; *Adv. Mater.* **21**, 1450–1472, 2009.).

As the reference diagram depicted below, when there is an interface dipole or a potential step at the interface, all the ionized electrons in the film will be affected by this interface potential step, so that their kinetic energy after leaving the film will change accordingly. As a result, the UPS spectrum will move as a whole. In this way, the changes in work function (Δ) or the VL shifts can be tracked by remeasuring the E_{cutoff} after depositing the organic films in sequence.

We have clarified the problem brought by the surface sensitivity of UPS in page 5 (the words in blue) of the revised manuscript. Although the change in work function can be captured in the thick bilayer film, the IP near the D-A interface and WF evolution with film thickness is unknown, i.e., there may be an abrupt shift at the heterojunction, or an extended gradient throughout most of the film. To solve that problem, we employed the Langmuir-Schäfer (LS) technique to build up the planar D-A heterojunction in monolayer precision, and we measured the IP and WF changes for each molecular layer in the vertical stack by UPS. It's precisely through this way that we now, for the first time, have mapped out the true energy level alignment for these complex donor-acceptor heterojunctions. This method has been successfully used to map the energy level bending in Inorganic/Organic interface before by our group (*Adv. Funct. Mater.* **26**, 1077–1084, 2016). Thanks to this nondestructive bottom-to-top method, we can map out the energy level alignment at the D-A interface.

We hope we have answered your concerns regarding the validity of the technique to determine the change in work function or VL shift, and we think we have provided strong enough evidence to support our claim in the manuscript.

Diagram 1. Schematic illustration of some of the important parameters derived from UPS characterization of surfaces and interfaces (Reference: *Adv. Mater.* **21**, 1450–1472, 2009)

Q2. Moreover, it is still not clear throughout the manuscript why the formation of interface dipole can drive effective exciton dissociation and charge generation in the case of negligible band offsets.

R2: Thanks for the reviewer’s comments. This is a valuable question, but the influence of energy level alignment on charge generation dynamics is beyond the scope of this manuscript, that is to map out the energy level alignment at the D-A interfaces in NFA-based systems with layer-by-layer precision. We demonstrate that the large VL shift exists at the interface which increases the interfacial state energy, and our findings reconcile the contradicting observations on energy offsets in literatures. We do mention the likely effect of the interface dipoles on charge generation mechanism mentioned on page 12 (the words in blue) with two references (*Appl. Phys. Lett.* **82**, 4605–4607, 2003; *Adv. Energy Mater.* **1**, 792–797, 2011). Although the latter reference is based on a fullerene system, both suggest that the existence of an interface dipole pointing from acceptor to donor side will facilitate the CT exciton dissociation into free charges and decrease the charge recombination. Nevertheless, further studies focused on the device physics to explore the relationship between the interface dipoles and charge generation mechanism in the NFA-based systems are needed, which is exactly what we are pursuing by using our LS-based fabrication approach. If successful, we certainly hope to present the work in the future.

Reviewer comments, second review -

Reviewer #1 (Remarks to the Author):

I read through the revised manuscript and response letter. The authors have done minor changes to the MS to provide further clarification on some of the issues raised, which have improved the manuscript. In my opinion, the results presented here are sufficiently novel and interesting to be published in Nat Comm. They provide significant additional insight into the energetic landscape at donor/non-fullerene acceptor heterojunctions as used in state-of-the-art organic PV devices, and the results go beyond those shown in previously published works. The conclusions are supported by the data. I could not identify further flaws in the experimental data and its analysis. The methodology applied appears sound to me and the work is meeting my expectations for a paper published in Nat Comm. Lastly, the Supporting Information provides significant additional data that further supports the claims of the work. I am supportive of publication as is.

Reviewer #2 (Remarks to the Author):

In the revised manuscript and response letter, the authors still did not give clear picture and evidence why the work function change can surely represent the vacuum level shift. In the present manuscript and supplemental materials, I did not see the UPS spectra like the Figure 2 in Adv. Mater. 21, 1450, (2009) to determine the cut off energy and work function in the planar bilayer D-A films. The yield of ionized electrons and their kinetic energy really depend on the band energy structure and distribution of electrons in the LUMO/HOMO and Fermi energy level, and it will critically affect the determination of the work function.

The authors claim that their findings reconcile the contradicting observations on energy offsets in literatures. But it looks to me that finding the small energy offset due to the interface dipole (large offset without interface dipole) does not really reconcile the contradicting observations. Only when it becomes clear that the interface dipole facilitates the charge separation though it leads to the small band offsets, one may claim the contradicting observations are reconciled.

Since the two major issues I raised in my last review report have not been clearly addressed, I cannot recommend the publication of the manuscript in its present form.

Response to Reviewers' Comments on Manuscript NCOMMS-21-26681A

We thank the reviewer for his or her helpful suggestions on improving this manuscript. We have carefully read the comments and have answered all the questions accordingly as detailed below.

Responses to Reviewer #2:

Comments to the Author:

In the revised manuscript and response letter, the authors still did not give clear picture and evidence why the work function change can surely represent the vacuum level shift. In the present manuscript and supplemental materials, I did not see the UPS spectra like the Figure 2 in Adv. Mater. 21, 1450, (2009) to determine the cut off energy and work function in the planar bilayer D-A films. The yield of ionized electrons and their kinetic energy really depend on the band energy structure and distribution of electrons in the LUMO/HOMO and Fermi energy level, and it will critically affect the determination of the work function.

The authors claim that their findings reconcile the contradicting observations on energy offsets in literatures. But it looks to me that finding the small energy offset due to the interface dipole (large offset without interface dipole) does not really reconcile the contradicting observations. Only when it becomes clear that the interface dipole facilitates the charge separation though it leads to the small band offsets, one may claim the contradicting observations are reconciled.

Since the two major issues I raised in my last review report have not been clearly addressed, I cannot recommend the publication of the manuscript in its present form.

Reply (R): We sincerely thank the reviewer for the careful examination and nice suggestions on our manuscript. We have answered the questions point by point as below and revised the manuscript accordingly.

Question (Q) 1. In the revised manuscript and response letter, the authors still did not give clear picture and evidence why the work function change can surely represent the vacuum level shift. In the present manuscript and supplemental materials, I did not see the UPS spectra like the Figure 2 in Adv. Mater. 21, 1450, (2009) to determine the cut off energy and work function in the planar bilayer D-A films. The yield of ionized electrons and their kinetic energy really depend on the band energy structure and distribution of electrons in the LUMO/HOMO and Fermi energy level, and it will critically affect the determination of the work function.

RI: Following the reviewer's suggestion, we put all the UPS spectra to determine the cutoff energy and work function in the planar bilayer D-A films in the supplementary information (**Supplementary Fig. 5**, in Supplementary information Pages 6–7), and mentioned it in the main text (Page 6, marked in red). The second question by the referee is quite profound and we address it as it as follows. The technique of using the secondary electron cut-off for determining the work function and hence the vacuum level position is well established (see

e.g., J. Phys.: Condens. Matter 20, 184008 (2008); Adv. Mater. 21, 1450 (2009); J. Phys. D: Appl. Phys. 50, 423002 (2017); for more details on UPS vacuum level-work function-Fermi level: Adv. Mater. 15, 271 (2003)). However, if one first measures the work function of e.g., a thick acceptor film, we obtain the position of the vacuum level in the surface layer of the film. If we then deposit a thick donor film on top and measure again, we obtain the position of the vacuum level in the surface layer of the donor film. If there is a difference in the vacuum level position between the acceptor surface and donor surface, we know we have a vacuum level shift, but that shift can be abrupt at the acceptor/donor interface or a linear gradient over the entirety of the donor film (see e.g., Org. Electron. 13, 1793 (2012)). This is precisely why the detailed Langmuir-Schäfer-based experiment is needed, where the UPS is carried out after each monolayer is deposited and the work function and hence vacuum level evolution can be tracked and mapped out (layer-by-layer) throughout the whole multilayer donor/acceptor and acceptor/donor stack.

Q2. The authors claim that their findings reconcile the contradicting observations on energy offsets in literatures. But it looks to me that finding the small energy offset due to the interface dipole (large offset without interface dipole) does not really reconcile the contradicting observations. Only when it becomes clear that the interface dipole facilitates the charge separation though it leads to the small band offsets, one may claim the contradicting observations are reconciled.

R2: We thank the reviewer for the comments. To address these comments, we will firstly briefly clarify the contradicting observation we claim to reconcile in our manuscript:

The contradicting observations on energy offsets in NFA-based systems:

Observation 1. Small or zero ΔHOMO or ΔLUMO for highly efficient donor-acceptor systems (Nat. Energy 1, 16089 (2016); Nat. Mater. 17, 119–128 (2018)): this energy offset is usually determined by the optical method ($E_g - E_{\text{CT}}$), in devices with abundant D-A interfaces, and a small/negligible $E_g - E_{\text{CT}}$ suggests a small/negligible ΔHOMO or ΔLUMO at the D-A interfaces. Assuming no energy level bending or interface vacuum level shifts, the results suggest that D-A systems featuring identical HOMO or LUMO levels can yield high-performance NFA OPVs.

Observation 2. Sizable ΔHOMO and ΔLUMO for systems that show small/negligible energy difference between E_g and E_{CT} (e.g., Nat. Mater. 20, 378–384 (2021) and others cited in the manuscript): the ΔHOMO and ΔLUMO are determined from neat D or A films by the PES (and IPES) methods. They hence observe that a large ΔHOMO and ΔLUMO energy offset before D-A contact, which is needed for high-performance NFA OPVs.

They are both correct observations but seemingly contradict each other, which can be reconciled by the missing piece to the puzzle: mapping the ELA at D-A interfaces. This is exactly one of our main contributions in this work. By mapping the ELA at D-A interfaces, we find that in high-performance NFA OPVs the interface dipole shifts the vacuum level leading to smaller ΔHOMO or ΔLUMO offset at the D-A interface (Observation 1, after D-A contact) compared to the large offset obtained from their corresponding neat films (Observation 2, before D-A contact). Indeed, in Nat. Mater. 20, 378–384 (2021) the authors realized that their observations were incompatible with the near zero $E_g - E_{\text{CT}}$ observed for the efficient NFA:donor systems. However, as their ΔHOMO values were inferred from HOMO values obtained from UPS measurement of neat films, not obtained at actual NFA/donor interfaces, a scheme featuring energy level bending due to the different molecular quadrupole

moments of the NFA at donors was proposed. Such energy level bending at organic donor:acceptor heterojunctions is indeed feasible, and can provide the necessary modification of the donor and acceptor energy levels in the vicinity of the interface while keeping a constant vacuum level as discussed in e.g., *J. Phys. Chem. Lett.* 3, 2374 (2012). This is then where one of our main contributions in this work is situated. By mapping the ELA at actual D-A interfaces with layer-by-layer precision, we find that in high-performance NFA OPVs there is no energy level bending (due to quadrupole shifts or any other mechanism), but instead, an interface dipole is formed that shifts the vacuum level, leading to small/negligible ΔHOMO (and ΔLUMO) offset at the D-A interfaces (Observation 1, after D-A contact) despite the large offset obtained from their corresponding neat films (Observation 2, before D-A contact).

We also believe our results are in agreement with Observation 2, i.e., a high ($\sim 0.4 - 0.5$ eV) before contact ΔHOMO is needed to ensure efficient free charge generation. We show that for such D-A combinations we obtain a significant interface dipole after D-A contact, see scheme 1, that has its positive side on the donor and negative side on the acceptor. Such interface dipoles are known from both theory (*App. Phys. Lett.* 82, 4605 (2003)) and experiment (*Adv. Energy Mater.* 1, 792 (2011)) to enhance the generation of free charges from excitons at organic D-A interfaces, thus in agreement with the papers featuring Observation 2.

We agree with the referee that carrying out an extensive study on the free charge generation and its dependence on interface dipole size, extension, D-A molecular orientation, D-A intermolecular distance, etc. would be highly interesting, but that must be a topic for a future separate manuscript.

Scheme 1. Energy levels of donor and acceptor before and after contact.

Reviewer comments, third review -

Reviewer #2 (Remarks to the Author):

I am glad to see that the authors have included the supplemental Fig. 5 to show the UPS spectra to determine the work function. I also strongly suggest to explain in the main text why the vacuum energy shift can be determined from the work function by referring to the same Fermi level in all the layers of the heterostructure during the whole growing process. It should be helpful to include the corresponding marks in Fig. 1 for better understanding. The authors also have given reasonable explanations why they claim to reconcile the contradictory observations. I recommend the publication in Nature Communications after revisions according to the above suggestions.

Reviewer #3 (Remarks to the Author):

The authors present use a very well control fabrication of planar heterojunctions to analyse the energy alignment at the D/A interface of two NFAs and two donor polymers. The effort here devoted to address the energy alignment in NFA/donor interfaces is recognized. The authors also measure by UPS (although the data is not presented) the energy alignment of the four compounds on substrates with different work function to obtain the pinning energies of the four compounds.

The most noteworthy observation is the VL shift observed when the acceptor layer (ITIC or IT4F) is deposited on top of the donor film on gold (D/Au), corresponding to an increase of the work function. Given that the observed change of VL cannot be reconciled with a scenario of Fermi level pinning to the LUMO of the acceptor (ICT-), the authors conclude the formation of an interfacial dipole. This conclusion seems to be well justified by UPS data. This implies a lower offset between HOMO levels as compared to the case of VL alignment.

The weakest part of the article is that a similar but opposite VL shift is not found when the donor layer is deposited on the A/AIOx. If the VL shift is a purely interfacial effect arising from the charge distribution at the interface, why the effect changes with the inverse deposition order, using AIOx as a substrate? This arises the question about the role of the substrate, whether the larger VL shift observed on D/Au may not be affected in a non-trivial way by the metal substrate.

This article presents a systematically performed experiment in mapping the energy alignment at the organic/organic interface. Still the article leaves some unclear questions in the discussion.

i) Role of the substrate: the substrate has a key role in the energy alignment of organic-organic heterostructures; it is not clear in this article about the choice of Au and AIOx as substrates. In terms of the organic-substrate interaction, a metal substrate is not equivalent to a non-metallic substrate, even when the work function is the same. It will be convincing if the authors demonstrate that similar interface dipole forms on ITO/ZnO substrate (used here for solar cells fabrication) or, at least, on other non-metallic substrates (AIOx, ITO, PEDOT:PSS...).

-The UPS spectra for the bare substrates should be included.

-A longer UPS spectrum should be shown in the supporting information to see deeper lying energy levels.

ii) NEXAFS: It is not obvious that NFA is face-on-oriented from the presented data. The NEXAFS spectra are not sufficiently explained.

iii) Interpretation and discussion: In a blend, D/A interfaces are randomly distributed in the space. Assuming that an interface dipole arises in some of the D/A interfaces having the negative charge in the acceptor side: is the local electric field not expected to be detrimental for the exciton dissociation?

iv) Figure 2 is rather confusing and the individual points are difficult to discern for each of the cases; it is recommended to present the results as energy diagrams, showing the change with thickness of work function and HOMO should facilitate the understanding.

It is a relevant topic. This article presents a systematically performed experiment in mapping the

energy alignment at the organic/organic interface. In the present form the article leaves however some doubts in relation to the interpretation.

Response to Reviewers' Comments on Manuscript NCOMMS-21-26681B

We thank the reviewers for the helpful suggestions on improving this manuscript. We have carefully read the comments and have answered all the questions accordingly as detailed below.

Responses to Reviewer #2:

Comments to the Author:

I am glad to see that the authors have included the supplemental Fig. 5 to show the UPS spectra to determine the work function. I also strongly suggest to explain in the main text why the vacuum energy shift can be determined from the work function by referring to the same Fermi level in all the layers of the heterostructure during the whole growing process. It should be helpful to include the corresponding marks in Fig. 1 for better understanding. The authors also have given reasonable explanations why they claim to reconcile the contradictory observations. I recommend the publication in Nature Communications after revisions according to the above suggestions.

Reply (R): We sincerely thank the reviewer for the helpful suggestions on improving our manuscript. Following the reviewer's comments, we complemented the description and explanation for determining the vacuum level (VL) and work function (WF) of multilayer films from UPS technique by referencing the same Fermi level, both in the main text (Page 4, words in red) and in the Supplementary information Fig. 1. We also improved the Fig.1 in the main text (Page 4) by adding ' E_F ' and 'WF' to mark the relationship between the VL shifts and WF changes.

Responses to Reviewer #3:

Comments to the Author:

The authors present use a very well control fabrication of planar heterojunctions to analyse the energy alignment at the D/A interface of two NFAs and two donor polymers. The effort here devoted to address the energy alignment in NFA/donor interfaces is recognized.

The authors also measure by UPS (although the data is not presented) the energy alignment of the four compounds on substrates with different work function to obtain the pinning energies of the four compounds.

The most noteworthy observation is the VL shift observed when the acceptor layer (ITIC or IT4F) is deposited on top of the donor film on gold (D/Au), corresponding to an increase of the work function. Given that the observed change of VL cannot be reconciled with a scenario of Fermi level pinning to the LUMO of the acceptor (ICT-), the authors conclude the formation of an interfacial dipole. This conclusion seems to be well justified by UPS data. This implies a lower offset between HOMO levels as compared to the case of VL alignment.

The weakest part of the article is that a similar but opposite VL shift is not found when the donor layer is deposited on the A/AlOx. If the VL shift is a purely interfacial effect arising from the charge distribution at the interface, why the effect changes with the inverse deposition order, using AlOx as a substrate? This arises the question about the role of the substrate, whether the larger VL shift observed on D/Au may not be affected in a non-trivial way by the metal substrate.

This article presents a systematically performed experiment in mapping the energy alignment at the organic/organic interface. Still the article leaves some unclear questions in the discussion.

i) Role of the substrate: the substrate has a key role in the energy alignment of organic-organic heterostructures; it is not clear in this article about the choice of Au and AlOx as substrates. In terms of the organic-substrate interaction, a metal substrate is not equivalent to a non-metallic substrate, even when the work function is the same. It will be convincing if the authors demonstrate that similar interface dipole forms on ITO/ZnO substrate (used here for solar cells fabrication) or, at least, on other non-metallic substrates (AlOx, ITO, PEDOT:PSS...).

-The UPS spectra for the bare substrates should be included.

-A longer UPS spectrum should be shown in the supporting information to see deeper lying energy levels.

ii) NEXAFS: It is not obvious that NFA is face-on-oriented from the presented data. The NEXAFS spectra are not sufficiently explained.

iii) Interpretation and discussion: In a blend, D/A interfaces are randomly distributed in the space. Assuming that an interface dipole arises in some of the D/A interfaces having the negative charge in the acceptor side: is the local electric field not expected to be detrimental for the exciton dissociation?

iv) Figure 2 is rather confusing and the individual points are difficult to discern for each of the cases; it is recommended to present the results as energy diagrams, showing the change with thickness of work function and HOMO should facilitate the understanding.

It is a relevant topic. This article presents a systematically performed experiment in mapping the energy alignment at the organic/organic interface. In the present form the article leaves however some doubts in relation to the interpretation.

Reply (R): We sincerely thank the reviewer for the careful examination and helpful suggestions on our manuscript. We have answered the questions point by point as below and revised the manuscript accordingly.

Question (Q) 1. Role of the substrate: the substrate has a key role in the energy alignment of organic-organic heterostructures; it is not clear in this article about the choice of Au and AlOx as substrates. In terms of the organic-substrate interaction, a metal substrate is not equivalent to a non-metallic substrate, even when the work function is the same. It will be convincing if the authors demonstrate that similar interface dipole forms on ITO/ZnO substrate (used here for solar cells fabrication) or, at least, on other non-metallic substrates (AlOx, ITO, PEDOT:PSS...).

-The UPS spectra for the bare substrates should be included.

-A longer UPS spectrum should be shown in the supporting information to see deeper lying energy levels.

RI: We thank the reviewer for pointing out this important issue. Yes, the substrates are important for the energy level alignment (ELA) in the organic-organic heterojunctions. In this work, we use high work function (WF) substrates (such as Au) to model the ELA in anode/D/A heterojunctions and use low WF substrates (such as Al/ AlO_x) to check the ELA in cathode/A/D heterojunctions. In this way, we can mimic the electrode/organic contact situation in efficient OPV devices where donors usually contact with an anode of high WF and acceptors usually contact with a cathode of low WF. Gold and Aluminum substrates are easy to prepare, feature stable WFs and are feasible for the LB/LS deposition method (e.g. PEDOT:PSS films are not as they are water soluble unless the PEDOT:PSS are cross-linked which changes the WF). To make these points clearer, we added more phrases to address the reason for the choices of substrates (in the main text, Pages 4–5).

To check the effect of metal vs non-metal substrates, we conducted extra UPS experiments on D-A bilayer films on metallic or non-metallic substrates to see if substrate/organic interaction would affect the interface dipoles at the D-A interface. As can be seen in Table 1 below, the relative larger D-A interface dipoles are still shown in PEDOT:PSS/D/A bilayer films compared to those in ZnO/A/D bilayer films. Similar D-A interface dipoles are measured in D/A bilayer films on PEDOT:PSS and Au substrates, and similar interface dipoles are also measured for A/D bilayer films on AlO_x and ZnO substrates. Thus, the substrates (metallic or non-metallic) do not play a crucial role in the cases studied in this work. However, the experiment results to exclude the influence from substrate species are necessary to be pointed out in the main text, so we added some phrases to address this concern (main text, Page 8, words in red) and put the Table below (Supplementary Table 4, Page 11) as well as the UPS spectra (bare substrates are included and deep levels are shown) in the supplementary information Fig. 8 (Pages 12–13), according to the reviewer's suggestions.

As for the reason why the sizes of interface dipoles are different in D-A films with the inverse deposition order, we tentatively attribute it to the different molecular packing modes at D-A interfaces from different deposition sequences and methods. We have pointed out this in the original manuscript (Page 8, words in blue), and we believe it would be another interesting topic in the future to discuss about the factors dominating the D-A interface dipoles systematically.

Table 1. Comparison of the interface dipoles in D-A bilayer films on different substrates.

Substrate	Au	AlO_x	PEDOT:PSS	ZnO
D-A interfaces	$\Delta\text{WF}_{\text{D/A}}$	$\Delta\text{WF}_{\text{A/D}}$	$\Delta\text{WF}_{\text{D/A}}$	$\Delta\text{WF}_{\text{A/D}}$
PBDB-T:ITIC	+0.60	-0.33	+0.71	-0.27
PBDB-T:IT4F	+0.65	-0.36	+0.57	-0.37
PM6:ITIC	+0.60	-0.16	+0.62	-0.24

Q2: NEXAFS: It is not obvious that NFA is face-on-oriented from the presented data. The NEXAFS spectra are not sufficiently explained.

R2: Thanks for the reviewer's careful examination and nice suggestions. Yes, the molecular orientations in spin-coated NFA films are more complicated to judge from NEXAFS spectra, compared to the Langmuir-Schäfer (LS) films, so we added a new table (Supplementary Table 1, page 5) to explain it in the supplementary information.

In the C K-edge NEXAFS spectra of spin-coated (SC) ITIC or IT4F films (supplementary Fig.4), also shown as below, the frontier structures from 284–286 eV are corresponded to the C 1s \rightarrow C=C π^* transitions, and the shoulder structures at the lowest photo energy around 284.4 eV are corresponded to the frontier edge of the LUMO structures of acceptors. Moreover, the frontier LUMO density of states is believed to be dominated by the end groups of the acceptors (A part in the A-D-A structure, shadowed with blue circle in the figure below), as the DFT calculation indicated in several literatures (J. Mater. Chem. A, 2018, 6, 23644; Spectrochim. Acta, Part A, 2021, 244, 118873). Thus, the orientation of the end groups in acceptors could be obtained from the spectral weight change of the shoulder peak at 284.4 eV with changing the incident angle, though the main C=C π^* transition peak at 285 eV do not give clear information.

From the NEXAFS spectra of ITIC-SC film, the end groups of ITIC show obvious face-on orientation where the shoulder peak at 284.4 eV shows higher intensity at grazing incident angle (20°), and lower intensity at normal incidence (90°). Furthermore, a near-planar configuration between the end groups and core structure is the most probable conformation owing to the alternating single-double bonds, so we think the average orientation of ITIC-SC film could be inferred from the obvious angular dependance of the end groups. However, we do not believe that the ITIC molecules present absolute planar configuration without any end-group rotations, thus we describe the orientation of ITIC-SC film carefully as 'preferentially face-on oriented', and the face-on orientation of ITIC films are also observed by another group (Chinese Phys. B 27, 2018).

However, it seems arbitrary to have described the orientation of IT4F-SC films simply as face-on, since the end-groups on IT4F do not show the same obvious angular dependance as ITIC-SC films. Thus, we revised the sentence as 'In contrast, ITIC films are preferentially face-on oriented, in agreement with reported results, whereas IT4F films are more disordered.' (main text, Page 5, Words in red).

Figure 1. Angular dependence of C K-edge NEXAFS spectra in total electron yield (TEY) detection mode for NFAs (ITIC, IT4F) made from spin-coating (SC) method, and the corresponding molecular structures of ITIC and IT4F.

Q3: Interpretation and discussion: In a blend, D/A interfaces are randomly distributed in the space. Assuming that an interface dipole arises in some of the D/A interfaces having the negative charge in the acceptor side: is the local electric field not expected to be detrimental for the exciton dissociation?

R3: Thanks for the reviewer's valuable questions. In this work, we demonstrate that, in NFA-based systems, interface dipoles exist at the D-A interface with their negative poles close to the acceptor side and positive poles close to the donor side. Instead of the likely detrimental effect of dipoles on the exciton dissociation, we believe that the dipole direction would be more favorable for the exciton dissociation and charge separation. Such interface dipoles are known from both theory (App. Phys. Lett. 2003, 82, 4605) and experiment (Adv. Energy Mater. 2011, 1, 792) to enhance the generation of free charges from excitons at organic D-A interfaces. What's more, similar opinions are held by the paper recently (Nat. Mater. 2021, 20, 378–384), where they demonstrate the interfacial bias potential (interface dipole observed in this current work) would facilitate the dissociation of CT states by simulation.

As mentioned by the reviewer, D/A interfaces are randomly distributed in the space, thus, to understand how the free charges are generated and transported through the randomly distributed D-A interfaces and pure domains, would be a super interesting but challenging work. Further studies focused on the device physics to explore the relationship between the interface dipoles and charge generation mechanism in the NFA-based systems are needed,

which is exactly what we are pursuing by using our LS-based fabrication approach. If successful, we certainly hope to present the work in the future.

Q4: Figure 2 is rather confusing and the individual points are difficult to discern for each of the cases; it is recommended to present the results as energy diagrams, showing the change with thickness of work function and HOMO should facilitate the understanding.

R4: Sincerely thank the nice suggestion from the reviewer. Following your advice, we redrew the Fig. 2 as energy diagrams for each D-A pairs (see below Version 3), showing the change of vacuum level (work function) and HOMO level (IP) with the film thickness or monolayer number. This figure shows each data points clearer, but unfortunately it seems harder for the audience to compare the IP values between different films, because one should calculate the IP value themselves by adding the WF and energy of valence band edge for each monolayer. However, we think the Version 1 (original version) would be better to deliver the important information more directly that abrupt shifts of the WF and no HOMO bending at the D-A interfaces. To solve the data overlapping problem, we divided the data points into two separated figures (Version 2). Here we attached all the three versions of Fig. 2, and we still think the original version is the most concise version which delivers the major information that we want to express.

Version 1 (the original version with minor revisions). All data are shown in one figure with direct WF and IP evolution information. It is good for comparison between different samples, but some data points are overlapped.

Fig. 2 Mapping the energy level alignment at the D-A interfaces in monolayer precision. IP and WF evolution of Au/D/A (solid lines) and AIO_x/A/D (dashed lines) heterojunctions as a function of D, A monolayer numbers (top axis) or the corresponding film thickness (bottom axis, presented by taking 3 nm as the average thickness of a monolayer). Upon the donor and acceptor stacking order, the substrate is in white, bottom phase is in red, and top phase is in blue.

Version 2. Two figures seperated from version 1 to avoid overlap of data points.

Fig. 2 Mapping the energy level alignment at the D-A interfaces in monolayer precision. IP and WF evolution of Au/D/A (solid lines) and AlO_x/A/D (dashed lines) heterojunctions as a function of D, A monolayer numbers (top axis) or the corresponding film thickness (bottom axis, presented by taking 3 nm as the average thickness of a monolayer). Upon the donor and acceptor stacking order, the substrate is in white, bottom phase is in red, and top phase is in blue.

Version 3. Energy level diagrams. More clear data points for each case are presented in eight separated panels, but direct information about WF and IP evolution is lost. Vacuum level and HOMO level are presented respected to the Fermi level. The audience need to calculate the IP (the distance between VL and HOMO) themselves.

Fig. 2 Mapping the energy level alignment at the D-A interfaces in monolayer precision. Vacuum level and HOMO level (both are referenced to E_F) of each monolayer are mapped versus the monolayer numbers (top axis) or the corresponding film thickness (bottom axis, presented by taking 3 nm as the average thickness of a monolayer). The left and right panel show the energy level evolutions in Au/D/A and AlO_x/A/D films, respectively.

Reviewer comments, fourth review -

Reviewer #2 (Remarks to the Author):

The authors have addressed all my comments and questions in the revised manuscript. I recommend publication of the paper in Nature Communications.

Reviewer #3 (Remarks to the Author):

I consider that the authors answered and addressed in a satisfactory way the questions. Just minor issues:

1) I agree with the authors that the Figure 2 is a good option . The so-called version 3 is very clear, I recommend its inclusion in the supporting information.

Note: remove the point of HOMO for thickness 0 nm.

2) Figure 2: what does it mean the IP value when there is not organic film, ie. thickness= 0nm?. With the given definition of IP, those point should be removed.

3) NEXAFS: Include a short explanation of the NEXAFs spectra in the SI (as that provided as response).

In contrast with the new paragraph in the p.5, in the article there is still appears a distinction between edge-on/edge-on and face-on/face-on (p.7, p.8, p.17). Please revise.

4) Provide the thickness of the films in Supplementary-Fig. 8 .

Response to Reviewers' Comments on Manuscript NCOMMS-21-26681C

We sincerely thank the reviewers for the nice suggestions on improving this manuscript. We have carefully read the comments and revised the manuscript accordingly as detailed below.

Responses to Reviewer #3:

Comments to the Author:

I consider that the authors answered and addressed in a satisfactory way the questions. Just minor issues:

1) I agree with the authors that the Figure 2 is a good option. The so-called version 3 is very clear, I recommend its inclusion in the supporting information.

Note: remove the point of HOMO for thickness 0 nm.

2) Figure 2: what does it mean the IP value when there is not organic film, ie. thickness= 0nm?.

With the given definition of IP, those point should be removed.

3) NEXAFS: Include a short explanation of the NEXAFs spectra in the SI (as that provided as response).

In contrast with the new paragraph in the p.5, in the article there is still appears a distinction between edge-on/edge-on and face-on/face-on (p.7, p.8, p.17). Please revise.

4) Provide the thickness of the films in Supplementary-Fig. 8.

Question (Q) 1. I agree with the authors that the Figure 2 is a good option. The so-called version 3 is very clear, I recommend its inclusion in the supporting information.

Note: remove the point of HOMO for thickness 0 nm.

Reply (R) 1: Thanks for the reviewer's nice advice. We have removed the point of HOMO for thickness 0 nm, added the revised figure (version 3) in the supplementary information (Supplementary Fig. 7, Page 11) and mentioned it in the manuscript (Page 6, words in red).

Q2. Figure 2: what does it mean the IP value when there is not organic film, ie. thickness= 0nm?.

With the given definition of IP, those point should be removed.

R2: Thanks for the reviewer's careful examination. Yes, we have removed the data points at 0 nm in Figure 2 (Main text, Page 7).

Q3. NEXAFS: Include a short explanation of the NEXAFs spectra in the SI (as that provided as response).

In contrast with the new paragraph in the p.5, in the article there is still appears a distinction between edge-on/edge-on and face-on/face-on (p.7, p.8, p.17). Please revise.

R3: Thanks for the reviewer's nice suggestions. We have included the explanation of the NEXAFS spectra in the supplementary information (Supplementary Note 1, Page 5).

In summary, all LS films of donors and acceptors studied in this work show obvious edge-on orientation, except that IT4F LS films are slightly disordered. Spin-coated donor films do not show any preferential orientation, which suggests a mixture of face-on and edge-on orientations in the films. Spin-coated acceptor films are preferentially face-on oriented except that IT4F films are less ordered. When fabricating the bilayer films with different deposition sequences and deposition methods, we can get the D-A interfaces with different orientations of face-on/face-on, face-on/edge-on, and edge-on/edge-on.

Following the reviewer's advice, we carefully checked and revised the description on the orientation of films from different methods. The revised sentences are shown as below.

Page 7: 'In this way we obtain a mixture of face-on/edge-on and edge-on/edge-on abrupt interfaces as the spin-coated films feature both face-on and edge-on molecular orientations.' is revised to 'In this way we obtain abrupt D-A interfaces with the top LS films featuring edge-on orientation (IT4F LS films are less ordered) and the bottom spin-coated films featuring both face-on and edge-on orientations.'

Page 8: 'The ELA displays near-identical behavior as the edge-on/edge-on with the VL shifts are of similar size and direction.' is revised to 'The ELA displays near-identical behavior as the LS/LS films of edge-on/edge-on orientation with the VL shifts are of similar size and direction.'

Page17: 'In summary, we provide direct measurement of the ELA at real D-A interface for NFA-based systems by building up both quasi and planar bilayers made from different methods, enabling different molecular orientations of face-on/face-on, face-on/edge-on, and edge-on/edge-on.' is revised to 'In summary, we provide direct measurement of the ELA at real D-A interface for NFA-based systems by building up both quasi and planar bilayers made from different methods, enabling different molecular orientations of preferentially face-on/face-on, face-on/edge-on, and edge-on/edge-on, as well as more randomly ordered interfaces.'

Q4. Provide the thickness of the films in Supplementary-Fig. 8.

R4: Yes, we have marked the thickness of films in this Figure, please see the Supplementary Fig. 9 in the revised version of Supplementary information (Pages 14–15).